# Bergamot (*Citrus bergamia*), a (Poly)Phenol-Rich Source for Improving Osteosarcopenic Obesity: A Systematic Review

**DOI:** 10.3390/foods13213422

**Published:** 2024-10-27

**Authors:** Giuseppe Mazzola, Mariangela Rondanelli, Giovanna Baron, Roberta Zupo, Fabio Castellana, Maria Lisa Clodoveo, Clara Gasparri, Gaetan Claude Barrile, Michela Seniga, Luca Matteo Schiavi, Alessia Moroni, Sukru Gulec, Patrizia Riso, Simone Perna

**Affiliations:** 1Endocrinology and Nutrition Unit, Azienda di Servizi alla Persona “Istituto Santa Margherita”, University of Pavia, 27100 Pavia, Italy; clara.gasparri01@universitadipavia.it (C.G.); gaetanclaude.barrile01@universitadipavia.it (G.C.B.); michela.seniga01@universitadipavia.it (M.S.); lucamatteo.schiavi01@universitadipavia.it (L.M.S.); alessia.moroni02@universitadipavia.it (A.M.); 2Department of Public Health, Experimental and Forensic Medicine, University of Pavia, 27100 Pavia, Italy; mariangela.rondanelli@unipv.it; 3Department of Pharmaceutical Sciences (DISFARM), University of Milan, Via Mangiagalli 25, 20133 Milan, Italy; giovanna.baron@unimi.it; 4Department of Interdisciplinari Medicine, University of Bari Aldo Moro, Piazza Giulio Cesare 11, 70100 Bari, Italy; roberta.zupo@uniba.it (R.Z.); marialisa.clodoveo@uniba.it (M.L.C.); 5Molecular Nutrition and Human Physiology Laboratory, Department of Food Engineering, Faculty of Engineering, İzmir Institute of Technology, Urla, 35430 Izmir, Türkiye; sukrugulec@iyte.edu.tr; 6Department of Food, Environmental and Nutritional Sciences, Division of Human Nutrition, University of Milan, 20133 Milan, Italy; patrizia.riso@unimi.it (P.R.); simoneperna@hotmail.it (S.P.)

**Keywords:** osteosarcopenia, sarcopenia, obesity, bergamot, *Citrus bergamia*, polyphenols

## Abstract

This systematic review investigates the potential of bergamot, a polyphenol-rich citrus fruit, in improving osteosarcopenic obesity, a condition characterized by the simultaneous presence of osteoporosis, obesity, and sarcopenia. Bergamot extracts have been suggested to possess several pharmacological properties, including anti-inflammatory and antioxidant effects, which could be useful in the management of age-related diseases and neuromuscular health. The review highlights the promising effects of bergamot extracts on skeletal muscle mass and function, particularly in the context of obesity, metabolic syndrome, osteosarcopenic obesity, and osteoporosis. Furthermore, some studies have shown that bergamot extracts can improve the metabolic balance, endothelial function, and maximal oxygen uptake in athletes, highlighting their potential benefits for skeletal muscle health. Taken together, these results suggest that bergamot extracts, especially those rich in polyphenols, may be a valuable adjunct in the management of osteosarcopenic obesity and other associated clinical conditions involving pro-inflammatory effects on organs and tissues.

## 1. Introduction

Skeletal muscle constitutes the largest tissue in the human body, accounting for approximately 40–50% of the total body mass of healthy individuals. It is crucial for maintaining metabolic health, particularly for glucose and insulin homeostasis [1,2]. Muscle mass refers to the amount of muscle tissue in the body, which constitutes a significant portion of total body weight. It is crucial for various physiological functions, including movement, metabolism, and overall health [3]. The significance of muscle tissue becomes increasingly apparent with aging. Muscle mass typically peaks during the third decade of life and begins to decline thereafter, particularly from the fourth decade onward. This decline is associated with various age-related diseases, including sarcopenia, which is characterized by a loss of muscle mass and strength. Maintaining muscle health is vital, as it serves as an independent predictor of mortality across all age groups, especially in older adults [1,2,3].

Many factors can influence muscle tissue mass and functionality. Some are endogenous, such as the circulating levels of certain hormones involved in muscle hypertrophy and anticatabolic processes (e.g., IGF1, GH, and testosterone); the level of subclinical chronic inflammation; insulin resistance conditions; genetic predispositions; the gut microbiome; and the gut–brain axis [3,4,5]. Others are exogenous and generally related to the exposome, primarily including the adoption of a healthy lifestyle based on proper nutrition, adequate physical activity, good sleep, hygiene, and effective stress management [3,4,5].

The peak and decline in muscle mass throughout life are influenced by several physiological factors. Type II muscle fibers, which are responsible for fast and powerful contractions, tend to atrophy more rapidly with aging, contributing to muscle mass loss. Additionally, IGF-1 (Insulin-like Growth Factor 1) signaling plays a critical role in muscle growth and regeneration, promoting protein synthesis and inhibiting protein degradation. The decline in IGF-1 levels with age further exacerbates muscle atrophy, leading to sarcopenia. IGF-1 (Insulin-like Growth Factor 1) promotes muscle growth by stimulating satellite cell proliferation and differentiation into mature muscle fibers, enhancing protein synthesis while inhibiting proteolysis (the breakdown of proteins) within muscles and mediating the effects of growth hormones on tissues, including the skeletal muscles [2,3,5].

Endogenous and exogenous factors associated with muscle aging are closely clinically and functionally interconnected. For example, insulin resistance conditions (strongly correlated with the adoption of a healthy lifestyle in a systemic sense) are often associated with an increase in subclinical chronic inflammation, which induces a decrease in skeletal muscle regenerative capacity and promotes various structural changes, such as an increase in fibrosis and a decrease in elastic and collagen fibers, which, in turn, lead to a significant decrease in muscle functionality and mass [3,4,5]. Among the diseases and/or pathophysiological conditions related to muscle health are neuromuscular diseases (e.g., spinal muscular atrophy, amyotrophic lateral sclerosis, and muscular dystrophy) and many diseases associated with connective and/or bone tissue alterations (e.g., osteoporosis, eosinophilic and diffuse fasciitis, polymyositis, etc.), metabolic alterations, especially of the lipid and glycemic profile (e.g., osteosarcopenic obesity, type 2 diabetes, and familial and non-familial dyslipidemia), age-related conditions (e.g., sarcopenia and neurodegenerative diseases like Parkinson’s, etc.), or those induced by physical trauma (e.g., athlete injuries) [3,6].

Osteosarcopenic obesity (OSO) is a syndrome marked by concurrent declines in bone density and muscle mass and an accumulation of excess adipose tissue, resulting in a diminished functional capacity and widespread metabolic dysregulation. This condition is of significant public health relevance due to its associations with elevated risks for immobility, falls, fractures, disability, and potentially other chronic diseases. Addressing OSO involves recognizing the interrelated nature of bone, muscle, and fat tissues, which requires comprehensive treatment plans that integrate nutrition and physical activity to mitigate the syndrome’s impact on the aging population [7]. In osteosarcopenic obesity (OSO), the intricate interactions between bone, adipose, and muscle tissues are significantly influenced by chronic inflammation, which disrupts homeostasis in these systems. Key inflammatory cytokines such as TNF-α and IL-6 play critical roles in this dysfunction. TNF-α contributes to muscle wasting by promoting apoptosis in muscle cells and inhibiting muscle regeneration, while also impairing osteoblast activity, leading to decreased bone formation. Similarly, IL-6, despite playing a protective role in acute inflammation, when chronically elevated, fosters muscle loss and enhances bone resorption by promoting osteoclastogenesis. This interplay creates a detrimental cycle where inflammation exacerbates both muscle and bone deterioration, underscoring the need for targeted therapeutic strategies that address these inflammatory pathways to improve outcomes for individuals suffering from OSO [7,8].

Bergamot is an endemic plant from the Calabria region in southern Italy whose fruit contains extremely high concentrations of bioactive compounds of considerable clinical interest [9,10]. The main bergamot polyphenols, particularly naringin and neohesperidin, have garnered attention for their roles in modulating inflammatory and antioxidant pathways, notably through the regulation of NF-κB and Nrf2 signaling. Naringin has been shown to inhibit NF-κB activation, which is crucial in mediating inflammatory responses, leading to a reduced expression of pro-inflammatory cytokines and mitigating inflammation-related tissue damage. Conversely, neohesperidin activates Nrf2, a transcription factor that promotes the expression of antioxidant enzymes, enhancing the body’s ability to combat oxidative stress [9,10,11,12,13]. However, when discussing plant extracts, it is essential to refer to the entire phytocomplex, which contains additional compounds that can interact positively or negatively with individual bioactive components. These interactions can enhance or diminish the therapeutic efficacy of the extract on biomolecular targets. This broader perspective on the phytocomplex underscores its potential not only for supporting muscle and bone health, but also for reducing chronic inflammation associated with metabolic disorders, making it a promising candidate for therapeutic interventions in conditions like osteosarcopenic obesity. Phytocomplexes extracted from different parts of bergamot, containing specific concentrations of polyphenols, have demonstrated various pharmacological properties that could potentially be very useful in the management of various aging-related pathologies and neuromuscular health [9,10]. Bergamot leaf extract (BLE) enhanced metabolic function, antioxidant capacity, and anti-inflammatory responses in the skeletal muscles in an experimental model of metabolic syndrome. Adorisio and colleagues and Sadeghi-dehsahraei and colleagues found that bergamot leaf extract showed positive metabolic effects and reduced oxidative stress and inflammation in skeletal muscles affected by metabolic syndrome or dyslipidemia [9,10].

Bergamot by-products have been identified as a sustainable source that can be used to counteract inflammation, suggesting their potential to improve skeletal muscle function [10]. Moreover, bergamot extracts have been shown to enhance metabolic function, antioxidant activity, and anti-inflammatory responses in the skeletal muscles in an experimental model of metabolic syndrome. Additionally, they offer benefits for the metabolic balance, endothelial function, and maximal oxygen uptake in athletes [14]. A literature review on the anti-inflammatory effects of bergamot in elderly people emphasizes its potential benefits for addressing inflammatory conditions like sarcopenia, particularly in older populations [11].

These studies collectively suggest the potential benefits of bergamot extracts in promoting and maintaining an adequate muscle mass and neuromuscular function in some clinical contexts, particularly in metabolic syndrome, obesity, and sarcopenia [9,10,11].

In this systematic review, we reported the main evidence regarding the use of bergamot extracts or polyphenols in promoting and maintaining an adequate muscle mass and counteracting obesity and osteoporosis, with a particular emphasis on the clinical management of osteosarcopenic obesity or pathologies associated with metabolic alterations, as well as age-related pathologies associated with alterations in bone and fat mass.

## 2. Materials and Methods

### 2.1. Search Strategy

A systematic literature search was conducted in the PubMed, Embase, and Cochrane databases from the 1 January 2000 to the present. The following search terms were used: (“bergamot” OR “*Citrus bergamia*”) AND (“skeletal muscle” OR “sarcopenia” OR “muscle mass” OR “muscle strength” OR “muscle function” OR “obese” OR “fat mass” OR “bone” OR “osteoporosis”). The following MeSH terms were used in the search strategy: Bergamot MeSH: *Citrus bergamia*. The reference lists of the included studies and pertinent review articles were manually examined to identify additional eligible studies. For further details, refer to Appendix A.

### 2.2. Study Selection

Two reviewers independently assessed titles and abstracts to identify studies of potential relevance. Full-text articles were retrieved and assessed for inclusion based on the following criteria: papers from 2000 to 2024, including in vitro, animal, and human studies in the English language.

### 2.3. Population

The systematic review included animal and human subjects across all age groups and related cellular models. The intervention comprised treatments with various forms of bergamot, including extract, juice, essential oil, and polyphenolic fractions. Comparators included either a placebo or the absence of any intervention. The specifics of the intervention involved the type of bergamot product used, the administered dose, and the duration of the treatment.

### 2.4. Primary Outcomes

The primary outcomes assessed included skeletal muscle mass, muscle strength, muscle function, and various biomarkers indicative of muscle and bone health. These outcomes were measured to evaluate the impacts of interventions involving bergamot extracts on muscle physiology and overall muscle health in OSO, obesity, and sarcopenic subjects.

Skeletal muscle mass, fat mass, and body mass evaluation: Skeletal muscle mass was assessed using imaging techniques such as dual-energy X-ray absorptiometry (DEXA), magnetic resonance imaging (MRI), and bioelectrical impedance analysis (BIA). These methods provided detailed insights into changes in muscle mass following the intervention with bergamot extracts. Body composition and visceral fat were also assessed by collecting anthropometric data such as hip and waist circumference and body weight assessment.

Muscle strength: Muscle strength was evaluated using standardized tests such as handgrip strength, isokinetic dynamometry, and one-repetition maximum (1RM) tests. These measures helped to determine the functional capacity of the muscles and the effectiveness of bergamot extracts in enhancing muscle strength.

Muscle function: Muscle function encompasses various performance metrics, including gait speed, balance tests, and endurance tests such as the six-minute walk test (6MWT) and timed up and go test (TUG). These tests helped to assess functional improvements in daily activities and overall mobility, providing a comprehensive view of how bergamot extracts influenced muscle functionality.

Biomarkers of muscle health and metabolic health: Biomarkers such as creatine kinase (CK), lactate dehydrogenase (LDH), C-reactive protein (CRP), and specific cytokines (e.g., IL-6 and TNF-α) were measured to evaluate muscle health and inflammation. The selected studies also investigated the roles of oxidative stress markers and antioxidant enzyme levels (e.g., superoxide dismutase [SOD] and catalase [CAT]) to understand the biochemical pathways influenced by bergamot extracts. Additionally, metabolic health biomarkers such as fasting glucose, insulin levels, lipid profiles (e.g., total cholesterol, LDL, HDL, and triglycerides), and insulin resistance markers (e.g., HOMA-IR) were included to provide a comprehensive picture of how bergamot extracts affected both muscle and metabolic health. These biomarkers helped to elucidate the mechanisms through which bergamot exerted its protective and restorative effects on skeletal muscle and metabolic functions.

### 2.5. Study Design

The studies considered in this review comprised randomized controlled trials (RCTs), pre-clinical studies involving animal models, and in vitro experiments. Any disagreements between reviewers were resolved through discussion or, when necessary, consultation with a senior third reviewer.

### 2.6. Study Selection and Data Extraction

The selection process entailed two independent reviewers screening titles and abstracts to identify relevant studies. Full-text articles were then retrieved and evaluated according to pre-defined inclusion criteria. Data extraction was performed by one reviewer and subsequently cross-verified by a senior reviewer to ensure accuracy. The extracted data included key study characteristics (author, year, country, and study design), participant information (age, sex, and health status), details of the intervention (type of bergamot product, dosage, and duration), and outcome measures.

### 2.7. Risk of Bias Assessment

In this systematic review, we evaluated the risk of bias using different tools tailored to the specific study types. For randomized controlled trials (RCTs), we applied the Cochrane criteria, which assess bias across the five following key domains: the randomization process, deviations from intended interventions, missing outcome data, outcome measurements, and the selection of reported results [15]. Pre-clinical animal studies were assessed using the Modified Cochrane Risk of Bias Tool, focusing on selection, performance, detection, attrition, and reporting biases. For in vitro studies, we used an adaptation of the SYRCLE’s risk of bias tool, which considers aspects such as test system characterization, protocol standardization, and result interpretation [16]. Disagreements were resolved by discussion.

## 3. Results

The literature search initially identified 512 articles. After a thorough screening process, 123 papers were selected for full-text review. Of these, 102 papers were excluded for not meeting the inclusion criteria, resulting in 21 studies being included in this systematic review, as illustrated in Figure 1.

We divided the results of these 21 studies into the following four main sections: skeletal muscle mass, obesity in humans, obesity in animals, and bone health.

### 3.1. Effect of Bergamot Extract on Skeletal Muscle

The results of the few studies found, overall, suggested that bergamot extracts may have positive effects on muscle health and physical performance in humans and animals.

Table 1 presents the studies examining the effects of bergamot extracts on skeletal muscles in humans and animals, while Figure 2 and Figure 3 illustrate the risk of bias assessments for the animal model studies and clinical trials, respectively.

#### 3.1.1. Effect of Bergamot Extract on Skeletal Muscles in Clinical Studies

Overall, the results from the clinical trials indicated promising preliminary results. Wolkodoff and colleagues [17] reported a significant improvement in VO2 max by 12% (*p* < 0.05, effect size = 0.8) and an enhancement in mood (Utian Quality of Life scores) by 15% (*p* < 0.05, effect size = 0.7). Mollace and colleagues [14] reported an increase in serum nitric oxide by 28% (*p* < 0.01, effect size = 0.5), an improvement in endothelial function by 22% (*p* < 0.01, effect size = 0.6), and an enhancement in VO2 max by 14% (*p* < 0.01, effect size = 0.7) compared to placebo, with no significant changes in heart rate (HR) or blood pressure (BP) at rest, but a lower HR at maximal exercise intensity by 8% (*p* < 0.05), suggesting that BPF Gold supplementation improves the ability of the blood vessels to dilate and constrict, which is critical for maintaining blood flow during exercise and an enhanced aerobic capacity. These findings suggest that bergamot extract supplementation can significantly enhance aerobic capacity and positively impact mental well-being and overall quality of life [14,17].

#### 3.1.2. Effect of Bergamot Extract on Skeletal Muscles in Animal Model Studies

The results from animal studies, though preliminary, provide evidence suggesting a positive impact on muscle mass. Palacio and colleagues [18] showed that BLE was able to reduce plasma leptin levels, improve insulin sensitivity, and attenuate oxidative stress and inflammation in the hypothalamus, adipose tissue, and skeletal muscles of rats fed with a high-sugar, high-fat diet. Specifically, in the skeletal muscles, BLE improved glucose uptake, elevated antioxidant enzyme activity, and reduced pro-inflammatory cytokine levels [18].

Overall, these studies seem to support the idea that bergamot extracts can positively impact physical performance by enhancing aerobic capacity, improving mood, and increasing nitric oxide levels.

### 3.2. Effect of Bergamot Extract on Obesity Subjects

Overall, the results from the selected obesity clinical studies and obesity animal models suggest that bergamot extracts may exert beneficial effects on fat mass, lipid profiles, and metabolic parameters in both humans and animals. The impacts of bergamot on obesity-related parameters in clinical and animal studies are detailed in Table 2 and Table 3, while Figure 4 and Figure 5 display the risk of bias assessments for the clinical trials and animal studies, respectively. In this section, we chose to present the findings by organizing them according to the type of bergamot extract used, given the large number of studies selected for this analysis. Priority is given to extracts that, based on their composition, appeared to offer the greatest therapeutic potential for osteosarcopenia. Studies involving bergamot in combination with other extracts are presented last, as they provide less direct evidence regarding the isolated effects of bergamot.

#### 3.2.1. Effect of Bergamot Extract on Skeletal Muscle in Clinical Studies

The clinical studies consistently indicate that bergamot extracts, particularly in their phytosomal form, exert significant effects on improving lipid profiles, as well as reducing body weight, BMI, and waist circumference in obese individuals. Furthermore, notable improvements in insulin sensitivity, along with reductions in markers of inflammation and hepatic steatosis, have been observed, underscoring the therapeutic potential of bergamot extracts in the management of obesity-related metabolic disorders.


**Bergamot Phytosomial Polyphenolic Fraction (BPF-P)**


Rondanelli et al. (2021) [19] conducted a randomized controlled trial (RCT) involving 80 overweight and obese individuals with a mean BMI of 27.86 ± 3.35 and a mean age of 59.03 ± 8.06 years. The participants received 150 mg of bergamot phytosome twice daily for 12 weeks. The study demonstrated significant reductions in body weight by 3.7%, BMI by 3.6%, and waist circumference by 3.4% (*p* < 0.001). Additionally, significant improvements in lipid profiles were observed, with total cholesterol being reduced by 21.4%, LDL by 24.6%, and triglycerides by 25.1%, while HDL levels increased by 6.8% (*p* < 0.01). No adverse effects were reported, suggesting that BPF phytosome is effective in improving metabolic parameters in obese individuals [19].


**Bergamot Polyphenolic Fraction (BPF) Not Phytosome and Bergamot Polyphenol Extract Complex (BPE-C)**


Several studies have investigated non-phytosomal BPF. Bruno et al. (2017) conducted two open-label, non-randomized studies on patients treated with second-generation antipsychotics (SGAs) [20,21]. In the first study, ref. [20], involving 28 patients, BPF (500 mg/day for 60 days) did not result in significant changes in body weight, BMI, or metabolic parameters, though 37.5% of patients showed some LDL cholesterol reduction [20]. In a follow-up study involving 15 patients treated with SGAs [21], a higher dose of BPF (1000 mg/day for 30 days) resulted in a 1.6% reduction in body weight (*p* = 0.004) and a trend of BMI reduction (*p* = 0.005), although no significant changes in other metabolic parameters were observed [21].

Gliozzi et al. (2014) [22] conducted an RCT with 107 patients with metabolic syndrome and NAFLD, administering 650 mg of BPF twice daily for 120 days. The study reported significant reductions in fasting plasma glucose by 17% (*p* < 0.05), LDL cholesterol by 37.7% (*p* < 0.05), triglycerides by 31% (*p* < 0.05), and an increase in HDL cholesterol by 28.9% (*p* < 0.05). Markers of hepatic steatosis, including ALT, AST, and γ-GT, also significantly decreased. Although there were no significant changes in body weight or visceral fat, BPF demonstrated notable improvements in lipid profiles and liver function [22].

Capomolla et al. (2019) [23] studied 52 obese patients (BMI > 26) administered with a bergamot polyphenol extract complex (BPE-C) for 90 days. The study showed significant reductions in fasting glucose (18.1%, *p* < 0.001), triglycerides (32%, *p* < 0.001), and cholesterol parameters (up to 41.4%, *p* < 0.001). Additionally, body weight decreased by 14.8% (*p* < 0.001) and BMI by 15.9% (*p* < 0.001), indicating a strong effect of BLE on improving metabolic parameters in obese individuals [23].


**Bergamot Extracts Combined with Other Extracts**


Ferro et al. (2020) [24] assessed the effects of a combination of bergamot polyphenolic fraction (BPF) and Cynara cardunculus (wild cardoon) extract in an RCT involving 102 subjects with NAFLD. The combination significantly reduced total cholesterol by 23.4%, LDL by 27.6%, triglycerides by 30.4%, and BMI by 3.5% (*p* < 0.001). Moreover, reductions in liver fat content and improvements in liver function markers (CAP score and hepatic steatosis markers) were observed, with a more significant effect in individuals with a BMI of ≥30 [24].

Hancke et al. (2021) [25] investigated a combination of BPF and Eurycoma longifolia in 97 obese participants. After 112 days of treatment, BMI was significantly reduced by 3.3% in the high-dose group and by 3.2% in the low-dose group (*p* < 0.001). However, no significant changes in lipid profiles or metabolic parameters were observed, indicating a limited effect on overall metabolic health compared to other formulations [25].

#### 3.2.2. Effect of Bergamot Extract on Skeletal Muscles in Animal Model Studies

Animal studies have demonstrated significant improvements in lipid profiles, insulin resistance, and inflammation with bergamot extracts, similar to the effects observed in human studies. However, the extent of metabolic improvements, particularly in body weight and triglyceride levels, tends to be greater in animal models compared to humans.


**Bergamot Polyphenolic Fraction (BPF) Not Phytosome**


Animal studies on non-phytosomal BPF have consistently shown significant improvements in metabolic parameters. Parafati et al. (2018) [26] evaluated the effects of BPF in C57BL/6J mice fed with a high-fat diet. Mice treated with 100 mg/kg/day of BPF for eight weeks exhibited an improved insulin sensitivity (HOMA-IR index) and a 30% reduction in blood triglycerides (*p* < 0.05). Additionally, hepatic inflammation decreased, as indicated by reductions in IL-6 mRNA levels and increases in anti-inflammatory IL-10 levels. BPF also reduced the total liver lipid content by 70% (*p* < 0.05) [26].

La Russa et al. (2019) [27] investigated BPF in Wistar rats fed with a cafeteria diet, with the results indicating that BPF (50 mg/kg/day for 14 weeks) significantly reduced body weight gain by 15%, triglycerides by 25%, and blood glucose by 20% (*p* < 0.05). The study also reported a 30% improvement in antioxidant capacity (*p* < 0.05), suggesting a broad impact on metabolic health in the context of diet-induced obesity [27].

Parafati et al. (2015) [28] conducted a similar study using Wistar rats and demonstrated that BPF treatment reduced body weight by 10% and fat mass by 15% (*p* < 0.01). The study also noted a 20% improvement in lipid profiles and a 25% improvement in glucose tolerance (*p* < 0.05), further confirming the metabolic benefits of BPF [28].

Mollace et al. (2011) evaluated BPF’s effects in both rats and humans. In Wistar rats, BPF administered at 10–20 mg/kg/day for 30 days resulted in significant reductions in total cholesterol by 30% (*p* < 0.001), LDL cholesterol by 40% (*p* < 0.001), and triglycerides by 35% (*p* < 0.01). These findings were paralleled by similar improvements in lipid profiles in human subjects with hyperlipidemia, suggesting the consistent efficacy of BPF across different models [12]. 


**Bergamot Leaves Extracts (BLE)**


Nakandakare-Maia et al. (2023) [29] studied the effects of BLE in Wistar rats fed with a high-sugar, high-fat diet. BLE (50 mg/kg/day) was administered for 10 weeks, leading to a 64% reduction in plasma leptin levels (*p* < 0.05) and a significant decrease in caloric intake (*p* < 0.05). BLE also reduced insulin resistance and oxidative stress in adipose tissue and the hypothalamus, showing potential benefits for overall metabolic health [29].

Siqueira et al. (2023) [30] conducted a control group, double-blinded, randomized study on 24 male Wistar rats divided into three groups (control, high-sugar–fat diet (HSF), and HSF + BLE). The rats were administered with 50 mg/kg/day of water-soluble BLE for 10 weeks. The study found that BLE significantly reduced triglyceride levels and insulin resistance while also lowering markers of inflammation and oxidative stress in hepatic and adipose tissues. Specifically, the pro-inflammatory markers TNF-α and IL-6 were reduced (*p* < 0.05), while antioxidant markers such as SOD and CAT were increased (*p* < 0.05). BLE also improved insulin sensitivity [30].

Siqueira et al. (2024) [31] performed a similar control group, double-blinded, randomized study on 40 male Sprague–Dawley rats divided into four groups (control, high-fat diet, high-fat diet + pomegranate extract low dose, and high-fat diet + pomegranate extract high dose). BLE was administered at 50 mg/kg/day for 10 weeks following a 20-week HSF diet. The study reported that BLE significantly reduced triglycerides by 35%, insulin by 22%, and insulin resistance by 34% (HOMA-IR, *p* < 0.001). BLE also reduced oxidative stress and inflammation markers in various tissues, including adipose tissues, heart, liver, and kidneys (*p* < 0.05 for all measured parameters) [31].


**Bergamot Extracts Combined with Other Extracts**


Musolino et al. (2020) [32] investigated Bergacyn^®^, a combination of BPF and Cynara cardunculus, in Wistar rats with non-alcoholic steatohepatitis (NASH). The study reported a 40% reduction in liver steatosis, a 50% reduction in inflammation, and a 15% reduction in liver weight (*p* < 0.05). Body weight was also reduced by 20%, indicating significant improvements in metabolic parameters associated with NASH [32].

Nucera et al. (2024) [33] evaluated the same combination in Sprague–Dawley rats fed with a high-fat diet. The study found that Bergacyn^®^ significantly reduced body weight gain by 15%, fat mass by 25%, and improved lipid profiles (total cholesterol reduced by 20% and LDL by 15%, *p* < 0.001). Additionally, markers of oxidative stress and liver damage were significantly improved, suggesting that combining bergamot extracts with other compounds may enhance their overall metabolic effects [33].

### 3.3. Effect of Bergamot Extract on Bone Mineral Density (BMD)

Overall, two of the analyzed studies suggested positive effects on bone structure and density and osteoclastogenesis [8,34]. Overall, both studies concluded that treatment with bergamot extracts can induce potentially useful biomolecular effects for the treatment of osteopenia, osteoporosis, and OSO conditions.

We selected no clinical studies regarding treatments with bergamot extracts and their effects on bone mass and bone mineral density (BMD). Table 4 shows the studies regarding the effect of bergamot extracts on BMD in animal models and in vitro studies, while Figure 6 and Figure 7 illustrate the risk of bias assessments for the animal model and in vitro studies, respectively.

#### 3.3.1. Effect of Bergamot Extract on Skeletal Muscles in Animal Model Studies

Only one study using wild-type C57/B6 mice and OPG knockout mice was selected for this review [34]. This study found that bergapten, a compound isolated from bergamot essential oil, significantly inhibits RANKL-RANK signaling transduction [34]. Bergapten suppressed the activation of the PI3K/AKT, JNK/MAPK, and NF-κB signaling pathways, which are involved in osteoclastogenesis. In addition, bergapten treatment protected trabecular bone structure, evidenced by a bone volume fraction (BV/TV) improvement of 22%, trabecular thickness (Tb.Th) improvement of 18%, and trabecular number (Tb.N) improvement of 20% [34]. Additionally, bergapten decreased osteoclastogenic differentiation by 19% and reduced systemic inflammation and cytokine production, including TNF-α, IFN-γ, IL-1, IL-1β, IL-2, IL-4, IL-6, IL-17, and IL-10, suggesting that bergapten exerts anti-inflammatory effects that contribute to its inhibition of bone resorption [34].

#### 3.3.2. Effect of Bergamot Extract on Skeletal Muscles in In Vitro Studies

To date, only one in vitro study using human osteoblast-like cell lines (Saos-2 and MG63) has been conducted [8]; in this study, treatments of Saos-2 and MG63 with bergamot polyphenol fraction (BPF) were shown to upregulate β-catenin (*p* = 0.001) and the osteoblast differentiation markers RUNX2 and COL1A. BPF downregulated RANKL (*p* = 0.028) without affecting osteoblast viability or proliferation. The study also found that BPF increased the protein levels of β-catenin and RUNX2 (*p* = 0.039) while reducing RANKL protein levels (*p* = 0.028), indicating a positive effect on osteoblast differentiation [8].

The results from both the animal and in vitro studies highlight the potential of bergamot-derived compounds, such as bergapten and BPF, in promoting bone health through the inhibition of osteoclastogenesis and the upregulation of osteoblast differentiation.

## 4. Discussion

This systematic review highlights the potential effects of bergamot extracts on bone mass, fat mass, and skeletal muscle mass and function, particularly in the context of insulin resistance and metabolic alterations, such as sarcopenia and osteosarcopenic obesity.

In the context of osteoporosis, bergamot extracts may contribute to maintaining bone density and reducing the risk of the condition by promoting bone formation and inhibiting bone resorption [8,34]. Chronic inflammation can activate the NF-κB pathway, which increases the production of pro-inflammatory cytokines like IL-6 and TNF-α. These cytokines can enhance osteoclast activity through the RANKL-RANK pathway, leading to increased bone resorption. Bergamot extracts, by reducing cytokine levels and modulating NF-κB signaling, may inhibit osteoclastic activity and promote bone formation through β-catenin activation and ERK1/2 phosphorylation, thus improving bone density and lowering the risk of osteoporosis [8,34].

In the context of skeletal muscle, fat adiposity, and metabolic health, although the studies selected for this review were limited in number and heterogeneous regarding dosage and treatment modalities, bergamot extracts appear to improve the metabolic activity in skeletal muscles by increasing glucose uptake and insulin sensitivity, which could contribute to a reduced risk of insulin resistance and metabolic syndrome, often associated with better clinical outcomes in osteosarcopenic obesity [22,23]. Furthermore, Rondanelli and colleagues [19] observed an association between a reduction in waist circumference and reductions in biomarkers associated with inflammation, such as TNF-α and IL-6, which are associated with a decline in chronic subclinical inflammation. These effects may slow the loss of muscle mass related to aging or the clinical course of osteosarcopenia and osteosarcopenic obesity, as observed in both clinical and pre-clinical studies [19,26,31]. A comparison of the different bergamot extracts examined in this review reveals that phytosomal formulations of BPF, such as those studied by Rondanelli et al. [19], tend to offer a greater bioavailability and more pronounced effects on metabolic parameters than non-phytosomal extracts [20,21,22]. Additionally, bergamot extracts rich in polyphenols, including those from leaves (BLE), demonstrate significant effects on insulin sensitivity and a reduction in oxidative stress, particularly in animal models [30,31]. However, these effects are often more potent in comparison to the outcomes observed with combinations of bergamot and other extracts, such as Cynara cardunculus, which, while effective in providing metabolic improvements, show more moderate results in terms of direct muscle and skeletal benefits [23]. Additionally, in one study it was reported that bergamot extracts could enhance endothelial function, which is crucial for maintaining blood flow and the oxygenation of muscles, thereby improving performance and muscle endurance, though these results were primarily demonstrated in athletes rather than in patients affected by osteosarcopenic obesity or related clinical conditions, such as obesity, sarcopenia, or osteoporosis [14]. In clinical conditions associated with metabolic syndrome, sarcopenia, and obesity bergamot extracts could contribute to optimizing body composition, potentially preventing the muscle mass loss associated with inflammation and insulin resistance more effectively than directly reducing fat mass, chronic inflammation, or muscle protein degradation [14,24,26,31].

The comparison of studies investigating the effects of bergamot extracts on skeletal muscles and metabolic health in animals versus humans reveals both similarities and notable differences, highlighting important translational implications.

In animal models, particularly those involving metabolic syndrome or obesity, bergamot extracts have consistently demonstrated positive effects on muscle mass, strength, and function. For instance, studies have shown that bergamot leaf extract can reduce oxidative stress and inflammation in skeletal muscles, leading to improved metabolic profiles and muscle performance. However, these studies often utilize high doses of bergamot polyphenols, which may not be directly translatable to human populations due to differences in metabolism and pharmacokinetics.

Conversely, human studies present a more complex picture. While some trials have reported improvements in muscle mass and function with bergamot supplementation, the results are less consistent than those observed in animal studies. Variability in dosing regimens, the duration of treatment, and participant characteristics (such as age and health status) contribute to these inconsistencies. Additionally, the doses used in human studies are generally lower than those employed in animal models, raising questions about their efficacy in achieving similar outcomes.

The translational implications of these findings suggest that, while animal studies provide valuable insights into the potential benefits of bergamot for enhancing muscle health and combating obesity-related conditions, the variability in human responses necessitates caution. Future research should aim for standardized dosing protocols and consider individual variability to better understand how bergamot extracts can be effectively utilized in clinical settings. Furthermore, bridging the gap between animal and human studies will require more comprehensive clinical trials that replicate the promising results seen in pre-clinical models while accounting for the complexities of human physiology.

The findings from this systematic review on bergamot extracts highlight significant practical implications for managing sarcopenia and osteosarcopenic obesity. The evidence suggests that bergamot, rich in polyphenols such as naringin and neohesperidin, can positively influence muscle mass and function while also addressing the metabolic dysregulation associated with obesity. These effects are particularly relevant for older adults, who are at an increased risk for sarcopenia and its complications. Given the promising results observed in both animal and preliminary human studies, there is a strong rationale for further clinical investigations to establish optimal dosing strategies and treatment regimens. From the results of the studies reviewed in this work, the ideal clinical target for potential treatment with bergamot extracts could be osteoporotic, sarcopenic, and OSO patients with chronic subclinical inflammation. It is also worth noting that, according to recent evidence, patients over 60 years old generally experience an increase in pro-inflammatory stimulation (“inflammaging”), which could justify the use of bergamot extracts even in the absence of overt inflammatory markers (e.g., elevated C-reactive protein) in this cohort of patients [35,36], particularly those with clinical conditions associated with alterations in inflammatory pathways and those related to OSO. In accordance with what has just been discussed, in certain contexts, the effects of bergamot might be limited. For individuals without metabolic problems or chronic inflammation, its beneficial effects may be less pronounced. Similarly, athletes with already optimized oxygenation and blood flow capacities might not derive significant additional benefits.

Despite the promising results highlighted in this systematic review, there are many methodological limitations, particularly concerning the heterogeneity of models and the variability in dosages used across different investigations. Many studies employed diverse experimental designs, including in vitro, animal, and human models, which complicates the comparison of results and the generalization of findings. Additionally, the doses of bergamot extracts administered varied widely, with some studies utilizing significantly higher concentrations than others, potentially leading to inconsistent outcomes regarding efficacy and safety.

Moreover, studies conducted with animal models and clinical trials used varying dosages, while some studies only evaluated combinations of bergamot with other phytochemical extracts [17,24,25,32,33]. This variability limits our ability to isolate which specific molecules or groups of molecules exert the observed biomolecular effects in clinical conditions related to metabolic, muscle, and bone health, such as osteosarcopenic obesity. Moreover, another methodological challenge observed in this review is the variability in the definition of study populations, which sometimes makes it difficult to compare findings across different clinical entities. For instance, in the study by Capomolla et al. (2019), obesity was not classified according to the international standard of a BMI of >30, which could limit the generalizability of their results [23]. Furthermore, in animal studies, the models used were not always obesity models. In several cases, to include a sufficiently large number of studies, a definition of obesity in animal models was employed that corresponded to a weight increase of more than 20% from the initial weight, rather than using more standardized models of obesity, as suggested by the National Research Council (US) Committee for the Update of the Guide for the Care and Use of Laboratory Animals (2011) [37]. These considerations introduce a degree of heterogeneity that must be considered when interpreting such findings for future clinical and pre-clinical research.

In summary, bergamot extracts seem to show a promising potential for improving muscle function and metabolic health, particularly relevant for patients with osteosarcopenic obesity, athletes, and individuals at risk of osteoporosis. However, further studies are needed to fully understand their mechanisms and determine their effectiveness in different contexts. Future research should be performed with well-designed randomized controlled trials that explore the long-term effects of bergamot supplementation on muscle health and metabolic outcomes in diverse populations, focusing on cohorts with uniform alterations in insulin sensitivity and inflammatory biomarkers, and standardized extraction methods, as well as dosing and treatment methodologies, should be employed to ensure the consistency and reliability of the results. Additionally, there is a lack of studies on how these effects are related to modifications in the gut microbiota and their role in modulating systemic inflammatory pathways driven by molecules such as short-chain fatty acids. Furthermore, the in vivo conjugation of bergamot phenolics is also another factor for its biological activation, since sulfates and glucuronides of hesperetin, naringenin, and eriodyctiol were observed in the serum of healthy volunteers [38]. Thus, the activity of bergamot should be evaluated regarding its conjugates. It is also useful to consider the potential synergies of bergamot extract with other treatments. The polyphenols present in bergamot extracts, such as neohesperidin, naringin, neoeriocitrin, brutieridin, and melitidin, have shown anti-inflammatory and antioxidant effects [11], which may complement the actions of other treatments for osteosarcopenic obesity, such as treatments that are focused on lowering cholesterol and diabetes. For instance, bergamot extracts could be used in combination with other natural compounds or medications that target insulin resistance, inflammation, or muscle function to enhance overall therapeutic outcomes. Also, bergamot contains bergamottin, a compound that inhibits the CYP3A4 enzyme, which is responsible for metabolizing many drugs [11]. This inhibition can lead to increased blood levels of certain medications, potentially causing adverse effects, and this can enhance the effects of statins, leading to muscle breakdown. Understanding these factors will enhance the clinical applicability of bergamot extracts as adjunct therapies in comprehensive management plans for sarcopenia and osteosarcopenic obesity, potentially improving quality of life and reducing the incidence of related health complications in aging populations.

## 5. Conclusions

In summary, bergamot extracts could be used in combination with other treatments for sarcopenic and osteosarcopenic obesity, leveraging the anti-inflammatory and antioxidant properties, especially of their polyphenols, to enhance overall therapeutic outcomes. From the overall analysis of studies included in this review, it seems that such treatments may be particularly useful in patients with insulin resistance and low-grade inflammation related to alterations in the redox balance and the uncontrolled activation of pro-inflammatory cytokine pathways, a very common comorbidity of sarcopenic and osteosarcopenic obesity related to a loss of muscle mass and muscle function, further highlighting the potential effects of these extracts on the muscle mass, bone mass, and fat mass of these patients. Nonetheless, the evidence reported appears to, overall, still be preliminary and characterized by a non-negligible degree of heterogeneity in the implemented extraction methods, dosages, and synergy with other phytochemicals of the bergamot extracts studied in the clinical, pre-clinical and in vitro studies included, which limits its current applicability in clinical contexts. Further research is necessary to fully understand the synergistic effects and optimal dosages for specific clinical contexts.

## Figures and Tables

**Figure 1 foods-13-03422-f001:**
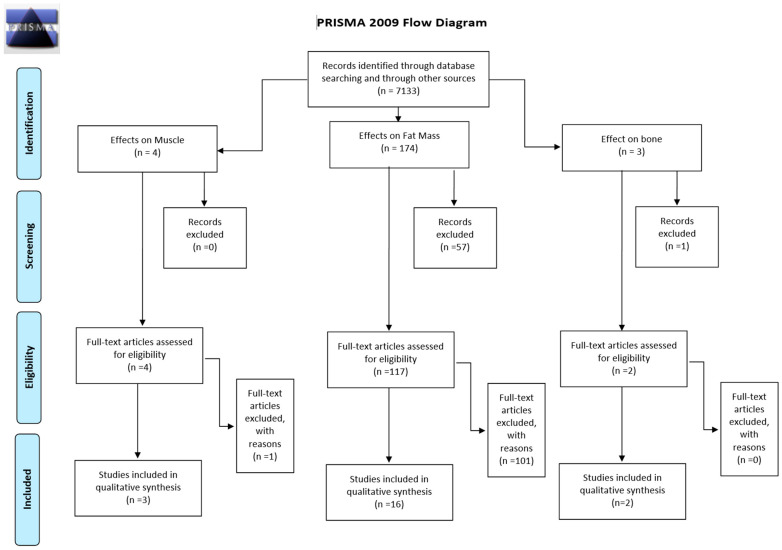
PRISMA flow diagram of studies selected.

**Figure 2 foods-13-03422-f002:**
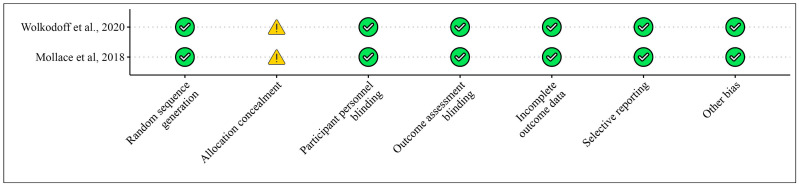
Bias for clinical studies included in the meta-analysis according to the Cochrane Risk of Bias Tool for Table 1. Bias designations by study criteria are indicated by 7 domains with categories including low risk if negative aspects of the study design were not likely to influence the study findings, high risk if the study design was likely to influence the study findings, or unclear risk if high or low risk could not be assigned because of a lack of evidence.

**Figure 3 foods-13-03422-f003:**
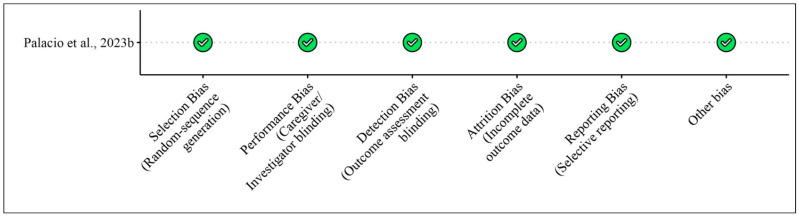
Risk of bias for pre-clinical studies (animal models) included according to the Modified Cochrane Risk of Bias Tool for Table 1. Bias designations by study criteria are indicated by 5 domains with categories including low risk if negative aspects of the study design were not likely to influence the study findings, high risk if the study design was likely to influence the study findings, or unclear risk if high or low risk could not be assigned because of a lack of evidence.

**Figure 4 foods-13-03422-f004:**
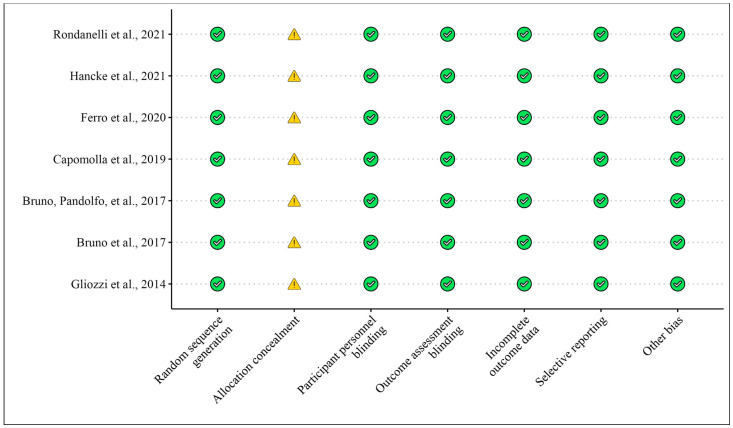
Risk of bias for clinical studies included according to the Cochrane Risk of Bias Tool for Table 2. Bias designations by study criteria are indicated by 7 domains with categories including low risk if negative aspects of the study design were not likely to influence the study findings, high risk if the study design was likely to influence the study findings, or unclear risk if high or low risk could not be assigned because of a lack of evidence.

**Figure 5 foods-13-03422-f005:**
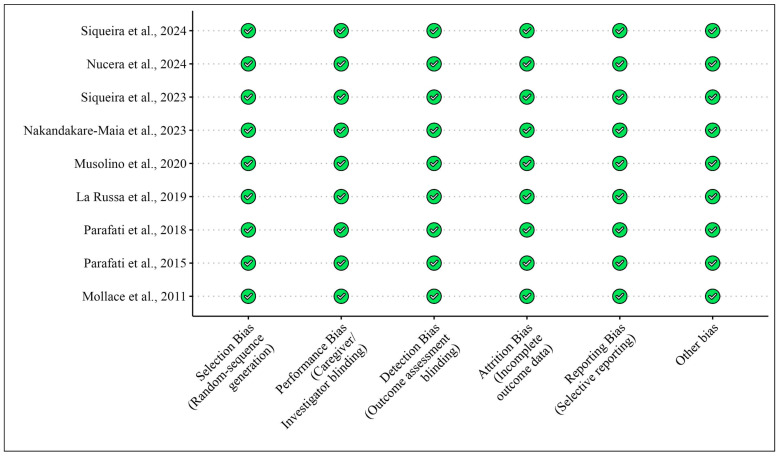
Risk of bias for pre-clinical studies (animal models) included according to the Modified Cochrane Risk of Bias Tool for Table 3. Bias designations by study criteria are indicated by 5 domains with categories including low risk if negative aspects of the study design were not likely to influence the study findings, high risk if the study design was likely to influence the study findings, or unclear risk if high or low risk could not be assigned because of a lack of evidence.

**Figure 6 foods-13-03422-f006:**
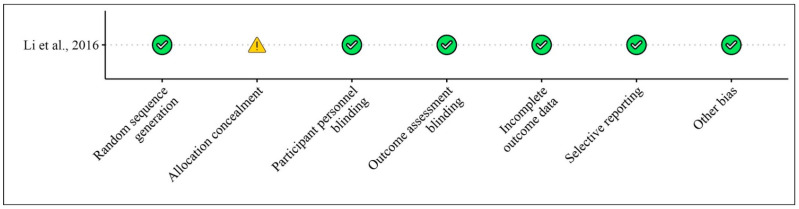
Risk of bias for clinical studies included according to the Cochrane Risk of Bias Tool for Table 4. Bias designations by study criteria are indicated by 7 domains with categories including low risk if negative aspects of the study design were not likely to influence the study findings, high risk if the study design was likely to influence the study findings, or unclear risk if high or low risk could not be assigned because of a lack of evidence.

**Figure 7 foods-13-03422-f007:**
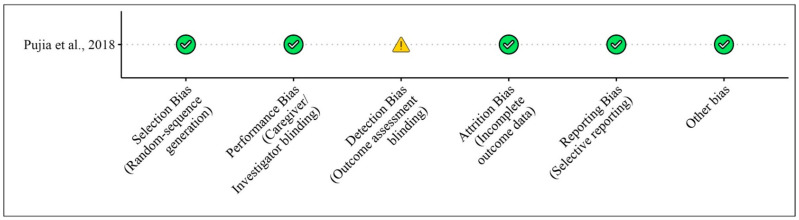
Risk of bias for in vitro pre-clinical studies included according to the SYRCLE’s risk of bias tool for Table 4. Bias designations by study criteria are indicated by 5 domains with categories including low risk if negative aspects of the study design were not likely to influence the study findings, high risk if the study design was likely to influence the study findings, or unclear risk if high or low risk could not be assigned because of a lack of evidence.

**Table 1 foods-13-03422-t001:** Effects of bergamot on skeletal muscles in humans and animals.

Paper	Type of Study and Level of Evidence	Compound/Extract	Sample	Posology/Treatment	Main Results
**Clinical Studies**					
Wolkodoff et al., 2020 [17]	RCT, Level 2	HerHeart (Bergamot supplement, water-soluble fraction, + Maca extract + Tribulus extract + Damiana extract, all titrated: 100 mg Bergamot, 50 mg Maca, 50 mg Tribulus, 50 mg Damiana per pill)	18 menopausal women, age: 62.5 ± 6.23, FM (fat mass): 13.90 ± 4.18 kg, FFM (fat free mass): 46.66 ± 5.55 kg	One pill twice per day for 60 days	Significant improvements in VO2 max by 12% (*p* < 0.05, effect size = 0.8) and mood (Utian Quality of Life scores) by 15% (*p* < 0.05, effect size = 0.7) compared to placebo group. No significant changes in body weight, body composition, or lean soft tissue.
Mollace et al., 2018 [14]	RCT, Level 2	Bergamot Polyphenolic Fraction (BPF Gold), water-soluble, containing 47% concentration of naringin, neohesperidin, neoeriocitrin, brutieridin, and melitidin, supplemented with 50 mg of ascorbic acid per dose	30 male cyclists, age: 28.1 ± 5.4, BMI: 24.4 ± 1.9	650 mg twice per day for 4 weeks; crossover with placebo after 2-week washout period	Increased serum nitric oxide by 28% (*p* < 0.01, effect size = 0.5), improved endothelial function by 22% (*p* < 0.01, effect size = 0.6), and enhanced VO2 max by 14% (*p* < 0.01, effect size = 0.7) compared to placebo. No significant changes in heart rate (HR) or blood pressure (BP) at rest, but lower HR at maximal exercise intensity by 8% (*p* < 0.05).
**Animal Studies**					
Palacio et al., 2023 [18]	Animal study with control group, randomized, Level 4	Bergamot Leaves Extract (BLE), titrated on flavonoids, 50 mg/kg/day, water-soluble, neoeriocitrin, naringin, neohesperidin (quantities not specified)	30 rats with metabolic syndrome induced by a high-fat diet, divided into 3 groups: Control (*n* = 10), Low Dose (*n* = 10), High Dose (*n* = 10)	50 mg/kg/day by gavage for 10 weeks; control and Metabolic Syndrome groups received vehicle (filtered water) by gavage daily; Metabolic Syndrome group received a high-sugar–fat diet (HSF)	Significant improvements in metabolic parameters, antioxidant activity, and reduced inflammation in skeletal muscle in both groups. Low-Dose group showed a 25% reduction in fasting blood glucose (*p* < 0.01) and a 30% increase in antioxidant enzyme activity (*p* < 0.05). High-Dose group showed a 35% reduction in fasting blood glucose (*p* < 0.01) and a 45% increase in antioxidant enzyme activity (*p* < 0.01). Biomarkers analyzed included fasting blood glucose, antioxidant enzyme activity (SOD, CAT), and inflammatory markers (TNF-α, IL-6).

**Table 2 foods-13-03422-t002:** Effects of bergamot on obesity (clinical studies).

Paper	Type of Study and Level of Evidence	Compound/Extract	Sample	Posology/Treatment	Main Results
**Bergamot Phytosomial Polyphenolic Fraction (BPF-P)**					
Rondanelli et al., 2021 [19]	RCT, Level 1	Bergamot phytosome (40% in weight BPF-P standardized to contain 11–19% total flavanones, containing neoeriocitrin, naringin, neohesperidin)	80 overweight and obese patients, BMI 27.86 ± 3.35, mean age 59.03 ± 8.06 years	150 mg bergamot phytosome twice daily for 12 weeks	Significant reduction in body weight by 3.7% (*p* < 0.001), BMI by 3.6% (*p* < 0.001), and waist circumference by 3.4% (*p* < 0.001). Improvements in lipid profile with reductions in total cholesterol by 21.4%, LDL by 24.6%, and triglycerides by 25.1% (*p* < 0.001). Increase in HDL by 6.8% (*p* < 0.01). No serious adverse effects reported. For subjects with BMI > 30 kg/m^2^, no specific subgroup analysis results were provided in the study.
**Bergamot Polyphenolic Fraction (BPF) Not Phytosome and Bergamot Polyphenol Extract Complex (BPE-C)**					
Bruno et al., 2017a [20]	Open-label, non-randomized, uncontrolled study, Level 3	Bergamot-Derived Polyphenolic Fraction (BPF), containing neoheriocitrin (55.535 mg), naringin (58.903 mg), neohesperidin (62.966 mg), melitidin (7.958 mg), brutieridin (24.371 mg), no excipients	28 outpatients on second-generation antipsychotics (SGAs), BMI 30.5 ± 7.55, 15 men and 13 women, mean age 45.8 ± 11.7	500 mg/day BPF for 60 days	BPF administration did not significantly change body weight (*p* = 0.849), BMI (*p* = 0.962), or metabolic parameters, including LDL, total cholesterol, HDL, triglycerides, and glucose (*p* > 0.05 for all). LDL reduction was minimal in 37.5% of patients. No adverse effects recorded.
Bruno et al. 2017b [21]	Open-label, non-randomized, uncontrolled study, Level 3	Bergamot-Derived Polyphenolic Fraction (BPF), containing neoheriocitrin (55.535 mg), naringin (58.903 mg), neohesperidin (62.966 mg), melitidin (7.958 mg), brutieridin (24.371 mg), no excipients	15 outpatients treated with second-generation antipsychotics (SGAs), BMI 35.0 ± 5.28, mean age 44.5 ± 9.1	1000 mg/day BPF for 30 days	BPF administration resulted in a statistically significant reduction in body weight by approximately 1.6% (*p* = 0.004) and a trend of BMI decrease by about 1.6% (*p* = 0.005). No significant differences in other metabolic parameters. One patient experienced adverse effects (heartburn).
Gliozzi et al., 2014 [22]	RCT, Level 1	Bergamot-Derived Polyphenolic Fraction (BPF), containing neoeriocitrin (370 ppm), naringin (520 ppm), neohesperidin (310 ppm), standardized at 38% polyphenols	107 patients with metabolic syndrome and NAFLD, BMI 29.4 ± 2.01, aged 56 ± 12 years	650 mg twice a day BPF for 120 days. Only overweight/obese individuals reached a 400–500 caloric restriction from the baseline intake	BPF significantly reduced fasting plasma glucose by 17% (*p* < 0.05), serum LDL cholesterol by 37.7% (*p* < 0.05), and triglycerides by 31% (*p* < 0.05), while increasing HDL cholesterol by 28.9% (*p* < 0.05). Significant reduction in hepatic steatosis markers (Steato test, ALT, AST, and γ-GT) and small, dense LDL particles by 35%. Hepatorenal index reduced significantly (*p* < 0.05). No significant reduction in body weight or visceral fat was reported. For subjects with BMI > 30 kg/m^2^, total cholesterol decreased by 25.7% (*p* < 0.01), LDL-C by 37.6% (*p* < 0.01), triglycerides by 31% (*p* < 0.01), fasting plasma glucose by 17% (*p* < 0.01), and hepatic steatosis markers by 45% (*p* < 0.01). The treatment resulted in a 35% decrease in small, dense LDL particles (*p* < 0.05) and improved hepatic function (ALT and AST reductions by 33% and 21%, respectively, both *p* < 0.05) and inflammatory markers (TNF-α by 25% and CRP by 22%, both *p* < 0.05).
Capomolla et al., 2019 [23]	RCT, Level 1	Bergamot Polyphenol Extract Complex (BPE-C), containing 80% BPE (bergamot juice extract with >8% pectins, >8% vitamin C), 20% BPFTM (bergamot juice flavonoids: 38 ± 2% neoeriocitrin, naringin, neohesperidin, melitidin, bruteridin)	52 obese patients with BMI > 26, atherogenic index of plasma (AIP) > 0.34, obesity with BMI > 26, high TG > 200 mg/dL, high TotChol > 200 mg/dL, moderate glycemia: 130 > Glu > 100 mg/dL), mean age for 650 mg 59 ± 7.6 and for 1300 mg 56.1 ± 10.9	650 mg and 1300 mg daily BPE-C for 90 days	BPE-C significantly reduced fasting glucose by 18.1% (*p* < 0.001), triglycerides by 32% (*p* < 0.001), cholesterol parameters by up to 41.4% (*p* < 0.001). AIP reduced to below 0.2 in the high-dose group (*p* < 0.001). Body weight decreased by 14.8% (*p* < 0.001) and BMI by 15.9% (*p* < 0.001) in BPE-C high-dose group. Reduced insulin levels and HOMA-IR. Significant reduction in circulating leptin (*p* < 0.001) and ghrelin (*p* < 0.001), and an increase in adiponectin (*p* < 0.001).
**Bergamot Extracts Combined with Other Extracts**					
Ferro et al., 2020 [24]	RCT, Level 1	Combination product (BC) containing bergamot polyphenolic fraction (BPFR) + wild type Cynara Cardunculus extract (CyC). Bergamot Polyphenolic Fraction (BPF) containing neoeriocitrin, naringin, neohesperidin, melitidin, brutieridin, exact percentages not specified. Excipients from both BC and placebo including PUFA and a mixture of bergamot pulp and albedo derivative	102 subjects with NAFLD (BMI mean: 28.9), mean age 51 ± 11	500 mg BPF twice daily for 60 days	BC significantly reduced total cholesterol by 23.4% (*p* < 0.001), LDL cholesterol by 27.6% (*p* < 0.001), triglycerides by 30.4% (*p* < 0.001), and increased HDL cholesterol by 11.3% (*p* < 0.001). Significant reduction in BMI by 3.5% (*p* < 0.001). Significant reduction in liver fat content (CAP score) and body weight in subjects with android obesity and BMI ≥ 30. Weight reduction: −3.0 ± 2 kg (placebo) vs. −4.5 ± 3 kg (BC), *p* = 0.005; CAP score reduction: −25.6 ± 46 dB/m (placebo) vs. −50.7 ± 40 dB/m (BC), *p* = 0.018; improvement in CAP: 44% (placebo) vs. 78% (BC), *p* = 0.007. CAP reduction more significant in those over 50 years. No adverse effects reported.
Hancke et al., 2021 [25]	RCT, Level 1	CitruSlim, a standardized blend of 83.33% Bergamonte *Citrus bergamia* Risso (fruit) extract (standardized to 38% polyphenols such as neoeriocitrin, naringin, neohesperidin, melitidin, brutieridin, 8% pectins, 8% vitamin C) and 16.67% Adapticort Eurycoma longifolia (root) extract (standardized to 22% eurypeptides, 40% glycosaponins)	97 obese participants, BMI 31.36 ± 2.56 for 200 mg 30.79 ± 2.33 for 400 mg, mean age 35.93 ± 9.56, and 35.90 ± 6.14;	200 mg (LD) or 400 mg (HD) CitruSlim daily for 112 days	CitruSlim HD and LD significantly reduced BMI by 3.3% and 3.2%, respectively (*p* < 0.001). No significant improvements in dyslipidemia or metabolic disturbances. Well-tolerated with no serious adverse effects reported.

**Table 3 foods-13-03422-t003:** Effects of bergamot on obesity (animal studies).

Paper	Type of Study and Level of Evidence	Compound/Extract	Sample	Posology/Treatment	Main Results
**Bergamot Polyphenolic Fraction (BPF) Not Phytosome**					
Parafati et al., 2018 [26]	Control group, double-blinded randomized, Level 4	Bergamot polyphenolic fraction (BPF), water-soluble, 40% flavonoids	40 male C57BL/6J mice, 12 weeks old, divided into four groups: control (*n* = 10), high-fat diet (*n* = 10), high-fat diet + extract low dose (*n* = 10), high-fat diet + extract high dose (*n* = 10)	8-week treatment, 100 mg/kg of body weight daily	BPF significantly improved insulin sensitivity (HOMA-IR index) and reduced blood triglycerides by 30% (*p* < 0.05). Reduced hepatic inflammation by decreasing Il-6 mRNA levels and increasing anti-inflammatory Il-10 levels (*p* < 0.05). Reduced total liver lipid content by 70% and improved NAS score (*p* < 0.05).
La Russa et al., 2019 [27]	Control group, double-blinded randomized, Level 4	Bergamot polyphenolic fraction (BPF), water-soluble, 40% flavonoids	26 male Wistar rats, 8 weeks old, divided into four groups: control (*n* = 5), control + BPF (*n* = 6), cafeteria diet (*n* = 7), cafeteria diet + BPF (*n* = 8)	14-week treatment, 50 mg/kg of body weight daily	Reduced body weight gain in cafeteria diet + BPF group by 15% (*p* < 0.05), decreased triglycerides by 25% and blood glucose levels by 20% (*p* < 0.05), improved antioxidant capacity by 30% (*p* < 0.05).
Parafati et al., 2015 [28]	Control group, double-blinded randomized, Level 4	Bergamot polyphenolic fraction (BPF), water-soluble, 40% flavonoids	28 male Wistar rats, 8 weeks old, divided into four groups: control (*n* = 7), control + BPF (*n* = 7), high-fat diet (*n* = 7), high-fat diet + BPF (*n* = 7)	10-week treatment, 50 mg/kg of body weight daily	Significant reduction in body weight by 10% and fat mass by 15% in high-fat diet + BPF group (*p* < 0.01), improved lipid profile by 20% and glucose tolerance by 25% (*p* < 0.05).
Mollace et al., 2011 [12]	Control group, double-blinded randomized, Level 2 (human) and Level 4 (animals)	Bergamot polyphenol fraction (BPF), water-soluble, 28% flavonoids	40 male Wistar rats, 8 weeks old, and 237 human patients with hyperlipidemia	30-day treatment, 10–20 mg/kg for rats, 500–1500 mg/day for humans	BPF treatment significantly reduced total cholesterol by 30% (*p* < 0.001), LDL cholesterol by 40% (*p* < 0.001), and triglycerides by 35% (*p* < 0.01) in rats. In patients, BPF reduced total cholesterol by 20–30% (*p* < 0.001), LDL by 23–38% (*p* < 0.001), and triglycerides by 28–41% (*p* < 0.001). For subjects with BMI > 30 kg/m^2^, no specific subgroup analysis results were provided.
**Bergamot leaves extracts (BLE)**					
Nakandakare-Maia et al., 2023 [29]	Control group, double-blinded randomized, Level 4	Bergamot leaves extract (BLE), water/ethanol (30/70% *v*/*v*), 50 mg/kg, not titrated	30 male Wistar rats, 8 weeks old, split into control diet (*n* = 10) and high sugar-fat diet (*n* = 20) groups	BLE treatment for 10 weeks, 50 mg/kg by gavage daily	BLE treatment significantly reduced plasma leptin levels in the HSF group by approximately 64% (*p* < 0.05). BLE treatment decreased caloric intake in the HSF group (*p* < 0.05). Reduced insulin resistance, oxidative stress, and inflammation in adipose tissue and hypothalamus (*p* < 0.05).
Siqueira et al., 2023 [30]	Control group, double-blinded randomized, Level 4	Bergamot leaves extract (BLE), water-soluble, titrated to contain polyphenols, 50 mg/kg/day	24 male Wistar rats, divided into 3 groups: Control (*n* = 8), high sugar-fat diet (HSF) (*n* = 8), HSF + BLE (*n* = 8), 20 weeks old	50 mg/kg/day by gavage for 10 weeks	BLE reduced triglyceride levels, insulin resistance, inflammation, and oxidative stress in hepatic and adipose tissues. It decreased levels of pro-inflammatory markers TNF-α (*p* < 0.05) and IL-6 (*p* < 0.05). Increased antioxidant markers such as SOD and CAT (*p* < 0.05). BLE improved insulin sensitivity.
Siqueira et al., 2024 [31]	Control group, double-blinded randomized, Level 2	Bergamot leaves extract (BLE), water/ethanol (30/70% *v*/*v*), 50 mg/kg, not titrated	40 male Sprague–Dawley rats, 10 weeks old, divided into four groups: control (*n* = 10), high-fat diet (*n* = 10), high-fat diet + pomegranate extract low dose (*n* = 10), high-fat diet + pomegranate extract high dose (*n* = 10)	50 mg/kg/day BLE for 10 weeks following 20 weeks of HSF diet	BLE significantly reduced triglycerides by 35%, insulin by 22%, and insulin resistance by 34% (HOMA-IR, *p* < 0.001). Decreased markers of oxidative stress and inflammation in adipose tissue, heart, liver, and kidneys (*p* < 0.05 for all measured parameters).
**Bergamot Extracts Combined with Other Extracts**					
Musolino et al., 2020 [32]	Control group, double-blinded randomized, Level 4	Bergacyn^®^ (Bergamot polyphenolic fraction (BPF) 50% + Cynara cardunculus (CyC) 50%), 200 mg/kg, liposoluble, titrated for catechins	24 male Wistar rats, 8 weeks old, divided into control group (*n* = 12) and treatment group (*n* = 12)	6-week treatment with polyphenol-rich extract, dosage not specified	Bergacyn^®^ significantly reduced steatosis by 40%, inflammation by 50%, and fibrosis in NASH (*p* < 0.02). Improved body weight by 20% and liver weight by 15% compared to controls (*p* < 0.05).
Nucera et al., 2024 [33]	Control group, double-blinded randomized, Level 2	Bergacyn^®^ (BPF 50% + CyC 50%), 200 mg/kg, liposoluble, titrated for catechins	36 male Sprague–Dawley rats, 10 weeks old, divided into three groups: control (*n* = 12), high-fat diet (*n* = 12), high-fat diet + extract (*n* = 12)	12-week treatment, 200 mg/kg of body weight daily	Bergacyn^®^ significantly reduced body weight gain by 15%, fat mass by 25%, and improved lipid profile (total cholesterol reduced by 20%, LDL by 15%), oxidative stress markers, and liver damage indicators. Significant reductions in eWAT weight (30%) and increased BAT weight (25%) (*p* < 0.001).

**Table 4 foods-13-03422-t004:** Results regarding the effects of bergamot extract on bone.

Paper	Type of Study and Level of Evidence	Compound/Extract	Sample	Posology/Treatment	Main Results
**Animal Studies**					
Li et al., 2016 [34]	Control group, double blinded-randomized, Level 4	Bergapten (5-methoxypsoralen, isolated from bergamot essential oil, other citrus essential oils, and grapefruit juice)	102 wild-type C57/B6 mice and 56 OPG knockout male C57/B6 mice, 6–8 weeks old	10 mg/kg/day and 20 mg/kg/day for 20 weeks	Bergapten significantly inhibits RANKL-RANK signaling transduction, suppressing the activation of PI3K/AKT, JNK/MAPK, and NF-κB signaling pathways (*p* < 0.01). It protects trabecular structure, evidenced by improved bone volume fraction (BV/TV) by 22% (*p* < 0.01), trabecular thickness (Tb.Th) by 18% (*p* < 0.01), and trabecular number (Tb.N) by 20% (*p* < 0.01). Additionally, it decreases osteoclastogenic differentiation by 19% (*p* < 0.01) and reduces systemic inflammation and cytokine production, including IL-2, IL-4, IL-6, IL-1, IL-1β, TNF-α, IFN-γ, IL-17, and IL-10. Bergapten treatment also prevents bone loss and improves bone mineral density by 15%, as seen in X-ray scans.
**In vitro studies**					
Pujia et al., 2018 [8]	In vitro study with human osteoblast-like cell lines, Level 5	Bergamot Polyphenol Fraction (BPF), titrated to contain 370 ppm neoeriocitrin, 520 ppm naringin, 310 ppm neohesperidin, water-soluble	Saos-2 and MG63 human osteoblast-like cell lines	0.001 mg/mL, 0.01 mg/mL, and 0.1 mg/mL for 24 h	BPF upregulates β-catenin (*p* = 0.001), osteoblast differentiation markers RUNX2 and COL1A, and downregulates RANKL (*p* = 0.028). It does not affect osteoblast viability or proliferation but increases protein levels of β-catenin and RUNX2 (*p* = 0.039), while reducing RANKL protein levels (*p* = 0.028). BPF also modulates ERK1/2 phosphorylation (*p* = 0.001).

## Data Availability

The data presented in this study are available on request from the corresponding author. The data are not publicly available due to privacy restrictions.

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
