# Peer review of "Bergamot (Citrus bergamia), a (Poly)Phenol-Rich Source for Improving Osteosarcopenic Obesity: A Systematic Review"

_foods, 2024, doi:10.3390/foods13213422_

Round 1
Reviewer 1 Report
Comments and Suggestions for Authors
This systematic review evaluates the health benefits of Bergamot in the context of osteosarcopenic obesity. The authors effectively outlined key factors that need to be considered to perform a systematic review of the efficacy of Bergamot, such as the specific part of the plant used, preparation methods, and study designs, ranging from in vitro and in vivo studies to human trials. However, these critical aspects are not adequately reflected in the study’s findings, leading to a more narrative review-like conclusion that does not fully capitalize on the strengths of a systematic review approach.
To enhance the review’s rigor and coherence, several areas require improvement:
- Descriptions of the test substances should be standardized, with clear details provided for the specific part of the Bergamot used (e.g., fruit, leaves), the method of preparation, and the key active components.
- References for each study should be explicitly listed in the table to ensure transparency and ease of verification.
- Since the review evaluates the effects of Bergamot on muscle, obesity, and bone separately, the PRISMA flow diagram (Figure 2) should be revised to clearly reflect the categorization and analysis of the 21 included studies according to these three distinct areas.
- Muscle studies (Table 1) and bone studies (Table 4) were grouped together across all study types, while obesity studies were divided into clinical studies (Table 2) and animal studies (Table 3). This inconsistency in presentation should be addressed to ensure uniformity across all sections of the review.
- As this is a systematic review, a totality analysis should be conducted for each category (muscle, obesity, and bone) to synthesize findings from all relevant studies to provide a comprehensive assessment of Bergamot’s efficacy.
Author Response
To enhance the review’s rigor and coherence, several areas require improvement:
- Descriptions of the test substances should be standardized, with clear details provided for the specific part of the Bergamot used (e.g., fruit, leaves), the method of preparation, and the key active components.
We have detailed in the tables which specific parts of the bergamot plant were tested, and where feasible, we have indicated the concentration of particular components (e.g., quantification of specific substances). It is important to note that it is not always possible to isolate specific compounds with pharmacological effects from a plant extract, as phytocomplexes often exert their actions through the relative concentrations of bioactive compounds. These compounds frequently act synergistically, and their interactions are not always fully understood.
- References for each study should be explicitly listed in the table to ensure transparency and ease of verification.
We have included citations in each table using the following format: "Author et al., year of publication (citation number)," ensuring that both the authors and publication year are visible alongside the numbered citation format as per the journal's guidelines. If the reviewers prefer a different format, we are open to making the necessary adjustments.
- Since the review evaluates the effects of Bergamot on muscle, obesity, and bone separately, the PRISMA flow diagram (Figure 2) should be revised to clearly reflect the categorization and analysis of the 21 included studies according to these three distinct areas.
The PRISMA flow diagram has been revised to incorporate the requested modifications, and the updated version is included in the revised manuscript. The diagram now clearly reflects the categorization and analysis of the 21 included studies according to the three distinct areas: muscle, obesity, and bone.
- Muscle studies (Table 1) and bone studies (Table 4) were grouped together across all study types, while obesity studies were divided into clinical studies (Table 2) and animal studies (Table 3). This inconsistency in presentation should be addressed to ensure uniformity across all sections of the review.
The tables on muscle mass (Table 1) and bone density (Table 4) present significantly fewer published studies compared to the data on obesity (excess fat mass). This is why we chose to subdivide the obesity-related evidence into animal studies and clinical studies, but not for muscle mass or bone density. Dividing the latter into separate tables for animal and clinical studies would result in creating a table for just a single publication. However, if the reviewer deems it more appropriate for the publication, we are willing to make this adjustment, although we suggest maintaining the current format for clarity and consistency.
- As this is a systematic review, a totality analysis should be conducted for each category (muscle, obesity, and bone) to synthesize findings from all relevant studies to provide a comprehensive assessment of Bergamot’s efficacy.
The selected evidence has already been categorized into the three areas mentioned (muscle, obesity, and bone), representing the totality of available studies retrieved from the PubMed database. We kindly ask if the reviewer could provide more detailed guidance on how they believe the study design could be further refined to conduct a more comprehensive systematic analysis that aligns with their expectations and optimally serves the research objectives.
Reviewer 2 Report
Comments and Suggestions for Authors
This is a topic that has been little studied and is relevant. However, the manuscript needs some adjustments. I hope that the suggestions below can contribute to improving the text and that these improvements can be implemented by the authors.
01 - The authors should clearly define central terms, such as "muscle mass", and better contextualize the importance of muscle tissue in aging. In addition, it would be interesting to better explain the physiology involved in the peak and decline of muscle mass throughout life, including the role of type II muscle fibers and IGF-1 signaling.
02 - The authors should better detail the interaction between bone, adipose and muscle tissues in OSO, including how inflammation affects bone and muscle homeostasis. Discussing the role of TNF-α and IL-6 in muscle and bone dysfunction deserves attention.
03 - The authors can detail and deepen the mechanisms of action of the main bergamot polyphenols, such as naringin and neohesperidin, in the regulation of inflammatory and antioxidant pathways (e.g., NF-κB, Nrf2).
04 - The methodology can be described in more detail
05 - I suggest reorganizing the text to group the results by biological mechanisms (e.g., reduced inflammation, improved insulin sensitivity, bone preservation), instead of by species (humans and animals).
06 - Detail the limitations of the reviewed studies, especially regarding the heterogeneity of the models and variability of the doses used.
07 - Compare and contrast the results of studies in animals and humans, discussing the translational implications.
08 - Discuss the results in terms of clinical relevance for sarcopenia and osteosarcopenia, exploring whether the observed effects are sufficiently robust to justify therapeutic interventions.
09 - Review the conclusion to emphasize the practical implications of the results, suggesting future research directions and clinical applicability.
Author Response
01 - The authors should clearly define central terms, such as "muscle mass", and better contextualize the importance of muscle tissue in aging. In addition, it would be interesting to better explain the physiology involved in the peak and decline of muscle mass throughout life, including the role of type II muscle fibers and IGF-1 signaling.
We have revised the introduction of the manuscript to include the requested clarifications. Specifically, we have provided a clear definition of key terms such as "muscle mass" and have contextualized the importance of muscle tissue in the aging process. Additionally, we have elaborated on the physiology of muscle mass changes throughout life, including the role of type II muscle fibers and IGF-1 signaling in the peak and decline of muscle mass.
02 - The authors should better detail the interaction between bone, adipose and muscle tissues in OSO, including how inflammation affects bone and muscle homeostasis. Discussing the role of TNF-α and IL-6 in muscle and bone dysfunction deserves attention.
We have revised the introduction to include a more detailed explanation of the interactions between bone, adipose, and muscle tissues in osteosarcopenic obesity (OSO). Specifically, we have discussed how inflammation affects bone and muscle homeostasis, emphasizing the roles of TNF-α and IL-6 in contributing to muscle and bone dysfunction. These pro-inflammatory cytokines are known to disrupt the balance of muscle and bone remodeling, exacerbating the pathophysiology of OSO.
03 - The authors can detail and deepen the mechanisms of action of the main bergamot polyphenols, such as naringin and neohesperidin, in the regulation of inflammatory and antioxidant pathways (e.g., NF-κB, Nrf2).
We have revised the introduction to include a more detailed explanation of the mechanisms of action of the main bergamot polyphenols, such as naringin and neohesperidin, in the regulation of inflammatory and antioxidant pathways. Specifically, we have expanded on their roles in modulating NF-κB and Nrf2 signaling. Naringin has been shown to inhibit NF-κB activation, reducing pro-inflammatory cytokine expression, while neohesperidin activates Nrf2, promoting antioxidant enzyme expression and enhancing the body's defense against oxidative stress. These pathways are critical in mediating the effects of bergamot polyphenols on inflammation and oxidative stress.
04 - The methodology can be described in more detail
To provide a more detailed description of the research methodology, we have included a supplemental appendix in both Excel and PDF formats. This appendix contains the Boolean queries used in the PubMed database, the total number of articles retrieved, and the number of articles selected after the screening process. This allows for full transparency and ease of reference for the reviewers.
05 - I suggest reorganizing the text to group the results by biological mechanisms (e.g., reduced inflammation, improved insulin sensitivity, bone preservation), instead of by species (humans and animals).
In the context of using plant extracts for the clinical management of complex conditions such as osteosarcopenic obesity, it is important to consider that individual alterations in biomolecular pathways, such as insulin sensitivity or subclinical chronic inflammation, are often interconnected and difficult to isolate as single variables. Consequently, most studies have assessed the effects of bergamot extracts in cohorts with specific physiological and/or pathophysiological conditions (e.g., obesity, sarcopenia, athletes) without further subdividing into smaller cohorts based on specific mechanisms (e.g., obese patients with insulin resistance). Therefore, to better reflect the nature of the available evidence, we chose to categorize the studies into animal models and clinical studies and describe the findings as reported by the authors of the included studies. However, in the future, as knowledge advances and with an higher number of studies conducted on humans and animals, the approach suggested by the reviewer could indeed be optimal for gaining a more in-depth understanding of the clinical applications of these findings and for establishing evidence-based medicine regarding the use of these extracts.
06 - Detail the limitations of the reviewed studies, especially regarding the heterogeneity of the models and variability of the doses used.
We have revised the discussion section of the manuscript to address the limitations of the reviewed studies, specifically highlighting the heterogeneity of the experimental models and the variability in the doses used. These factors contribute to challenges in comparing results across studies and may affect the generalizability of the findings.
07 - Compare and contrast the results of studies in animals and humans, discussing the translational implications.
We have revised the discussion section to include a comparison between the results of animal and human studies, highlighting the key similarities and differences. We have also discussed the translational implications of these findings, emphasizing how animal models can provide valuable insights but may not always fully replicate human physiology. This comparison underscores the need for cautious interpretation when extrapolating results from animal studies to clinical applications in humans.
08 - Discuss the results in terms of clinical relevance for sarcopenia and osteosarcopenia, exploring whether the observed effects are sufficiently robust to justify therapeutic interventions.
We have revised the results section to discuss the clinical relevance of the findings for sarcopenia and osteosarcopenia. We explored whether the observed effects are sufficiently robust to justify therapeutic interventions, noting that while the results are promising, further clinical trials with standardized protocols and larger sample sizes are needed to confirm the efficacy of bergamot extracts in treating these conditions. The current evidence suggests potential benefits, but additional research is necessary to establish clear therapeutic guidelines.
09 - Review the conclusion to emphasize the practical implications of the results, suggesting future research directions and clinical applicability.
We have revised the conclusion to emphasize the practical implications of the results, highlighting the potential clinical applicability of bergamot extracts in managing conditions like sarcopenia and osteosarcopenia. Additionally, we have suggested future research directions, including the need for well-designed clinical trials, standardized dosing regimens, and investigations into the long-term safety and efficacy of these extracts. These steps are crucial for translating the current findings into evidence-based therapeutic interventions.
Round 2
Reviewer 1 Report
Comments and Suggestions for Authors
I had expected the compound/extract section in each table to be systematically organized, but there has been no change. For example, in Table 2a, entry #19 refers to a fruit juice, entry #20 is a polyphenol fraction from an unspecified part, and entry #23 is a mixture of a polyphenol fraction and other substances. Entry #26 appears to be an extract from leaves using 70% ethanol, but the formatting is inconsistent, making it difficult for readers to easily understand these distinctions. Furthermore, when evaluating the efficacy of Bergamot, if different parts and extraction methods are used, they cannot be considered the same substance. Therefore, a justification is needed for how these were combined in the systematic review.
The research question of this study is designed to include different types of studies, allowing the results from those various study designs to be combined. However, each study type has different designs and objectives. Clinical trials focus on human subjects and assess real-world effects, while animal and in vitro studies are more focused on understanding basic mechanisms. Therefore, it is crucial to clearly distinguish between them and analyze their results in a balanced way.
Finally, in this research, the effects on muscle, fat mass, and bone were studied separately, with a particular emphasis on fat mass, where many clinical trials have been conducted. Therefore, at least for this area, in addition to bias analysis, it is necessary to provide a quantitative synthesis report. Please refer to Reproductive Biology and Endocrinology, 2022; 20: 176.
Author Response
Thank you for your suggestions regarding the organization of the results and discussion sections.
We have carefully revised these sections, as well as the tables, to reflect your comments as much as possible. Specifically, we have made efforts to systematically organize the compound/extract entries to clarify the differences in the substances, their parts, and extraction methods. Additionally, we have improved the separation of study types (clinical, animal, and in vitro) and provided a more structured presentation of their findings.
We have also reorganized the results into multiple sections to enhance clarity. In the discussion, we have included a comparison of the outcomes obtained from different types of extracts, in addition to the previously existing comparisons between cellular models, animal models, and clinical studies, as you suggested.
All these changes are highlighted in yellow within the text to make your review easier and more efficient.
The primary outcomes assessed in this systematic review include various metrics, for example related to muscle mass, strength, and function, which may not be directly comparable. This variability can lead to difficulties in calculating a standardized effect size for inclusion in a funnel plot, as you suggest in this section of your answer: “quantitative synthesis report … Please refer to Reproductive Biology and Endocrinology, 2022; 20: 176….” In addition, this review indicates that some studies had a risk of bias that can distort effect sizes and undermine the validity of the findings, further complicating the creation of a reliable funnel plot.
We trust that these revisions enhance the clarity and rigor of the manuscript.
Reviewer 2 Report
Comments and Suggestions for Authors
The authors have complied with the review requests and improved the manuscript. I assess that the manuscript is now qualified. Therefore, I suggest that the current version be accepted for publication.
Author Response
my sincere thanks to the reviewer for their valuable suggestions and insights, which have greatly contributed to improving the quality of the manuscript. I am grateful for the time spent reviewing it and for accepting the manuscript